# Emerging Imaging Technologies in Forensic Medicine: A Systematic Review of Innovations, Ethical Challenges, and Future Directions

**DOI:** 10.3390/diagnostics15111410

**Published:** 2025-06-01

**Authors:** Feras Alafer

**Affiliations:** College of Applied Medical Sciences, Jouf University, Sakaka 72388, Saudi Arabia; aferf@ju.edu.sa

**Keywords:** forensic imaging, artificial intelligence, virtual autopsy, operational challenges, ethical considerations, interdisciplinary collaboration, cultural sensitivity

## Abstract

Forensic medicine has increasingly integrated advanced imaging technologies to improve the accuracy and efficiency of investigations. Techniques such as virtual autopsy, multi-detector computed tomography (MDCT), and artificial intelligence (AI)-driven imaging have revolutionized the identification of injuries and causes of death. Despite these advancements, the field faces operational, ethical, and legal challenges that hinder widespread adoption. **Objectives**: This systematic review aimed to explore the role of emerging imaging technologies in forensic medicine, identify key challenges in their implementation, and provide insights into optimizing their use in forensic practice, with particular attention to cultural, ethical, and interdisciplinary aspects. **Methods**: A systematic review was conducted following PRISMA guidelines. Literature searches were performed across six databases, including PubMed, Scopus, Web of Science, and others, focusing on studies discussing imaging technologies in forensic contexts. A total of **10 studies** were included after applying eligibility criteria. The data were synthesized using narrative synthesis and thematic analysis. **Results**: Four key themes emerged: (1) advancements in AI and imaging technologies, (2) operational and financial barriers, (3) ethical and legal considerations, and (4) interdisciplinary collaboration and training. Emerging imaging modalities enhance diagnostic precision and facilitate non-invasive examinations, offering culturally sensitive alternatives to traditional autopsies. However, high costs, algorithmic biases, data security risks, and the lack of standardized forensic imaging protocols present significant challenges. The potential for cross-cultural and international forensic collaborations through AI-enabled imaging was also identified as a promising future direction. **Conclusions**: Advanced imaging technologies hold transformative potential in forensic medicine. Addressing financial, ethical, and operational challenges through interdisciplinary collaboration, standardized guidelines, and culturally sensitive practices is crucial for maximizing their utility and global acceptance

## 1. Introduction

Forensic medicine is a specialized field at the intersection of healthcare and the legal system, charged with applying medical knowledge to criminal and civil cases [1,2]. Over the past few decades, the global demand for accurate, efficient, and timely forensic evaluations has intensified, driven in part by increasing caseloads, evolving legal standards, and advancements in medical technology [3,4]. Key to this field is the ability to identify causes and mechanisms of injury or death, which not only aids in legal adjudication but also contributes to public health measures aimed at reducing preventable fatalities [5]. Within this context, medical imaging has emerged as an indispensable tool, offering non-invasive, detailed insights into anatomical structures and pathological changes that are essential for forensic investigations [6].

Medical imaging techniques ranging from conventional X-rays to state-of-the-art modalities such as computed tomography (CT), magnetic resonance imaging (MRI), and micro-computed tomography (micro-CT) have profoundly transformed forensic examinations [7]. These imaging technologies enable forensic specialists to detect subtle fractures, hemorrhages, and other forms of trauma that may otherwise remain hidden during external or even internal examinations [8,9]. In addition, advanced imaging can preserve visual evidence digitally, supporting re-analysis and serving as robust documentation for court proceedings [10]. For instance, virtual autopsy, or “virtopsy”, which combines CT or MRI scans with 3D reconstruction, provides a comprehensive view of the deceased’s internal state without the need for invasive dissection [11]. Rather than replacing traditional autopsies, virtopsy methods act as complementary tools that enhance conventional practices by offering non-invasive options where conventional autopsy may be limited by cultural, religious, or ethical considerations [12].

Despite these promising developments, the implementation of advanced medical imaging in forensic practice presents numerous challenges. One significant hurdle is the high cost associated with state-of-the-art imaging equipment and the specialized training required for operators and interpreters [13,14]. Many regions, particularly in low-resource settings, face severe budgetary constraints that limit their ability to invest in such technology, thereby widening disparities in the quality and speed of forensic investigations [15,16]. Even in well-resourced jurisdictions, the operational costs of running CT or MRI machines—along with the need for meticulous calibration and frequent maintenance—can strain existing healthcare and forensic systems [17]. Moreover, although these imaging modalities can capture an impressive amount of data, the shortage of professionals qualified to interpret such complex images can introduce delays and inconsistencies in forensic diagnoses [18,19].

Another layer of complexity arises from the ethical, legal, and cultural considerations associated with forensic imaging. Digital storage and transmission of highly sensitive post-mortem images risk potential breaches of privacy if not managed with strict cybersecurity measures [20]. Additionally, the admissibility of imaging findings as legal evidence can be contested if protocols are not standardized or if concerns arise regarding the chain of custody and data integrity [21]. These concerns highlight the urgent need for developing robust guidelines and standard operating procedures to regulate all stages of forensic imaging, from image acquisition to secure storage and court presentation [22,23].

Importantly, cultural and religious sensitivities exert a profound influence on the acceptance of postmortem practices, particularly in Islamic countries, parts of the Middle East, and among certain Christian and Jewish communities [24]. Traditional autopsies are often met with resistance in these contexts due to beliefs surrounding bodily integrity after death [25]. Virtual autopsy techniques, by offering non-invasive alternatives, provide a culturally respectful solution that upholds both forensic requirements and religious norms. Furthermore, these technologies open opportunities for international scientific collaboration [26]. Imaging data can be securely transmitted across jurisdictions, enabling experts from different countries, cultures, and religious backgrounds to jointly examine forensic cases without physically disturbing the deceased. This cross-cultural model of forensic investigation holds the potential to standardize forensic practices globally while respecting diverse sociocultural values [27].

Equally critical is the need for interdisciplinary collaboration between forensic pathologists, radiologists, anthropologists, and data scientists. Historically, forensic pathology and radiology have operated somewhat independently, with forensic pathologists focusing on autopsies and radiologists working primarily in clinical domains [28]. However, the complexity of modern forensic cases now demands integrated teamwork. Radiologists must deepen their understanding of forensic-specific injuries and artifact interpretation, while forensic pathologists must become proficient in utilizing advanced imaging modalities [29]. Structured interprofessional training programs and seamless communication channels are essential to ensure that the benefits of imaging technologies are fully realized in forensic practice [30].

The emergence of artificial intelligence (AI) and machine learning (ML) further promises to revolutionize forensic imaging. Algorithms trained on large datasets of forensic images can assist in detecting trauma patterns, estimating postmortem intervals, and even predicting mechanisms of injury [31]. These tools could significantly reduce subjective variability in image interpretation and accelerate diagnostic workflows, which is crucial for time-sensitive forensic investigations [32]. However, concerns about algorithmic bias, data privacy, and legal defensibility must be rigorously addressed to ensure responsible integration of AI technologies into forensic practice [33].

Given the growing reliance on imaging technologies and the complex sociocultural, operational, and ethical challenges they present, this systematic review aims to critically evaluate the role of emerging imaging technologies in forensic medicine. It further seeks to identify opportunities for optimizing their implementation through standardized protocols, interdisciplinary education, and culturally sensitive approaches that promote global forensic collaboration.

### 1.1. Aim of the Study

This systematic review aims to critically evaluate and synthesize the current literature on innovative medical imaging techniques within forensic medicine and elucidate the challenges and limitations that may hinder their widespread adoption. By providing a comprehensive overview, this review offers actionable insights for forensic practitioners, healthcare administrators, and policymakers, thereby facilitating the development of robust, standardized practices that leverage imaging technologies for more accurate and efficient forensic diagnoses.

### 1.2. Research Question

How do emerging imaging technologies influence diagnostic accuracy and efficiency in forensic medicine, and what are the primary operational, ethical, and interdisciplinary challenges that impact their successful implementation?

## 2. Materials and Methods

### 2.1. Search Strategy and Selection Criteria

This systematic review was conducted in accordance with the Preferred Reporting Items for Systematic Reviews and Meta-Analyses (PRISMA) guidelines to ensure methodological rigor, transparency, and reproducibility. A detailed research protocol was developed and prospectively registered with the International Prospective Register of Systematic Reviews (PROSPERO) under the registration number CRD42024625274, reflecting our commitment to systematic integrity and scholarly accountability.

To ensure a comprehensive identification of relevant literature, a systematic search strategy was employed across multiple leading databases, including PubMed, Scopus, Web of Science, CINAHL, Embase, ProQuest, and Elsevier/ScienceDirect. The final database search was executed on 1 September 2024, encompassing publications from database inception through the search date. The search strategy utilized a combination of Medical Subject Headings (MeSH) and free-text keywords to capture the breadth of research involving forensic medicine, medical imaging technologies, artificial intelligence applications, and operational and ethical challenges in forensic practice. Search terms were iteratively refined in consultation with subject experts to maximize sensitivity and specificity for the topic of forensic imaging innovations. Key terms included but were not limited to: “forensic medicine”, “post-mortem imaging”, “virtual autopsy”, “computed tomography”, “magnetic resonance imaging”, “artificial intelligence”, “machine learning”, “diagnostic accuracy”, “operational barriers”, “ethical challenges”, and “interdisciplinary collaboration”. Boolean operators (AND, OR) and truncation were used to optimize each search across databases, tailored to the indexing conventions of each platform. The detailed search strategy is presented in Table 1.

Studies were included if they evaluated any form of medical imaging technology ranging from conventional radiography to advanced modalities such as computed tomography (CT), magnetic resonance imaging (MRI), and emerging technologies like micro-CT or artificial intelligence applications in the context of forensic medicine. We focused on identifying both quantitative and qualitative research examining diagnostic innovations, limitations, and interdisciplinary challenges.

### 2.2. Eligibility Criteria for Screening

After the initial removal of duplicate records, the eligibility screening process commenced with a comprehensive two-stage procedure: first, a title and abstract review; followed by full-text assessment of articles that met the preliminary criteria. The screening process was designed to ensure methodological rigor and adherence to the predefined inclusion and exclusion criteria.

This systematic review targeted peer-reviewed empirical studies, systematic reviews, scoping reviews, and technical reports that investigated the application of advanced imaging technologies, including artificial intelligence (AI)-enhanced modalities, in forensic medicine. Eligible studies included those reporting on diagnostic performance, ethical or legal considerations, operational challenges, and interdisciplinary collaboration relevant to the adoption of these technologies in forensic practice.

Studies were considered eligible if they:Examined the use of imaging modalities such as computed tomography (CT), magnetic resonance imaging (MRI), micro-CT, or virtual autopsy techniques in forensic contexts.Reported on AI or machine learning applications in postmortem imaging or forensic diagnostics.Discussed challenges (e.g., high costs, algorithmic bias, legal admissibility, training deficits) or enabling factors (e.g., digital infrastructure, cross-cultural collaboration, standardization) related to imaging technology implementation in forensic medicine.Were published in English, involved human subjects or cadaveric studies, and included clearly defined methodologies and outcomes relevant to forensic diagnostics.

Both qualitative and quantitative studies were included, provided they offered substantial insight into the integration, impact, or implementation of advanced imaging technologies in real-world forensic settings. Studies presenting interdisciplinary, sociocultural, or legal perspectives were also included if they directly related to postmortem imaging applications.

Exclusion criteria were applied to:Case reports, narrative commentaries, editorials, letters to the editor, and conference abstracts lacking methodological detail.Studies focusing exclusively on live clinical imaging (e.g., diagnostic imaging in living patients) or on non-imaging forensic methods such as bloodstain pattern analysis, toxicology, or DNA analysis.Papers that did not address forensic imaging, lacked empirical findings, or failed to report challenges, facilitators, or measurable outcomes related to imaging integration in forensic investigations.Non-English publications without available translations.

From an initial pool of 3892 records identified across database searches, 980 duplicates were removed, leaving 2912 unique records for title and abstract screening. Following this initial screening, 2183 articles were excluded for not meeting basic inclusion criteria (e.g., lack of imaging relevance, theoretical nature, or clinical-only focus). 729 full-text articles were subsequently retrieved and assessed in detail for eligibility. Of these, 719 articles were excluded for reasons including non-imaging focus (n = 290), conceptual-only content (n = 247), insufficient forensic relevance (n = 113), and lack of methodological clarity (n = 69). Ultimately, 10 studies met all inclusion criteria and were selected for final synthesis. A PRISMA flow diagram (Figure 1) is provided to transparently illustrate the screening and selection process, including reasons for exclusion at each stage [16,34,35,36,37,38,39,40,41,42].

### 2.3. Data Extraction Process

Data extraction was a pivotal phase in this systematic review, designed to systematically compile and synthesize relevant information from the selected studies examining the use of advanced imaging technologies and artificial intelligence (AI) in forensic medicine. The primary objective of this phase was to extract critical data to illuminate the innovations, challenges, and implementation dynamics, shaping the integration of these technologies into forensic investigative practice.

The data extraction process involved a detailed and structured review of each included article, guided by a standardized data extraction template. The process focused on the following core components:**Study Characteristics**: Key information was extracted, including study design (e.g., systematic review, scoping review, narrative review, or empirical analysis), publication year, country or region of the study, and forensic setting (e.g., forensic pathology units, radiology departments, or medicolegal institutes). Where applicable, sample size and population demographics (e.g., types of forensic cases, professional roles involved) were also documented to provide contextual understanding and assess the transferability of findings.**Imaging Technologies and AI Innovations**: Data were collected on the imaging modalities studied, such as computed tomography (CT), magnetic resonance imaging (MRI), post-mortem angiography, micro-CT, and emerging AI-enabled image analysis techniques. We noted the intended purpose of each technology—whether for injury detection, time-of-death estimation, cause-of-death analysis, or virtual autopsy applications. Where available, details were also recorded about the stage of implementation (pilot, research-only, or institutional use) and the specific software or algorithmic models employed.**Challenges and Limitations**: A major focus of extraction involved identifying and categorizing the challenges and limitations reported in each study. These were grouped under operational (e.g., cost, infrastructure), technical (e.g., data interpretation complexity, image resolution), legal (e.g., admissibility in court, data custody), ethical (e.g., privacy of digital postmortem data), and professional training barriers (e.g., lack of radiological expertise among forensic pathologists). Emphasis was placed on contextual factors that influenced technology adoption in different legal and cultural settings.**Outcomes and Implementation Insights**: We extracted reported outcome measures including diagnostic improvements (e.g., injury detection accuracy, reduction in subjective interpretation), workflow efficiency, cultural acceptability, and impact on legal proceedings. Studies offering insight into strategies for implementation—such as interdisciplinary training programs, protocol standardization, or cross-border forensic collaborations—were noted for.

### 2.4. Quality Assessment

To evaluate the methodological rigor of each included study, we utilized the ROBIS (Risk of Bias in Systematic Reviews) tool, adapted to assess the quality of both primary research and secondary reviews focusing on forensic imaging. This structured approach helped to identify potential biases in study design, data analysis methods, and result interpretation. Two reviewers independently assessed each study for risk of bias, focusing on aspects such as clarity of research questions, appropriateness of methodology, transparency in reporting, and whether confounding variables were adequately addressed. Any disagreements between reviewers were resolved through a consensus-based discussion, ensuring that the final quality ratings were consistent and objective.

### 2.5. Data Analysis

In this systematic review examining emerging imaging technologies and artificial intelligence (AI) applications in forensic medicine, data analysis was conducted using a structured, multi-dimensional approach that combined **narrative synthesis** and **thematic analysis**. This dual-method strategy enabled a comprehensive exploration of both the practical applications and the operational, ethical, and legal implications associated with integrating advanced imaging in forensic settings. The analytical framework was designed to accommodate the diversity of methodologies among included studies and to distill both descriptive insights and conceptual patterns relevant to the implementation and evolution of forensic imaging technologies.

#### 2.5.1. Narrative Synthesis

Narrative synthesis served as the foundational component of the analytical process. This method facilitated a systematic comparison and contextual interpretation of findings across heterogeneous study designs and settings, including forensic institutes, academic hospitals, and medico-legal laboratories. The synthesis enabled us to categorize and describe the various imaging modalities utilized in forensic medicine such as computed tomography (CT), magnetic resonance imaging (MRI), micro-CT, and virtual autopsy and to examine the ways in which AI-enhanced tools (e.g., trauma detection algorithms, postmortem interval estimators, automated injury classification) were applied in forensic diagnostics. In addition, this approach supported the exploration of interdisciplinary workflows and outlined how emerging technologies were received by different professional stakeholders, including pathologists, radiologists, and legal practitioners.

Through narrative synthesis, attention was paid to the operational contexts in which these technologies were deployed, variations in diagnostic accuracy, and the degree to which the tools addressed sociocultural and ethical considerations, particularly in regions where invasive autopsies may be culturally restricted. Findings were also synthesized in terms of real-world feasibility, data integrity safeguards, and the utility of AI tools for standardizing forensic reporting.

#### 2.5.2. Thematic Analysis

In parallel with narrative synthesis, a thematic analysis was conducted to systematically identify and explore recurring themes and concepts across the included studies. This qualitative technique enabled a deeper examination of underlying issues and contextual patterns, beyond what was observable through descriptive summary alone. A manual coding framework was applied to the extracted data and refined iteratively through peer discussion to ensure thematic rigor and conceptual clarity.

Emergent themes included:**Technological advancements and diagnostic precision**: The impact of AI and high-resolution imaging on improving forensic accuracy and efficiency.**Operational and resource-related barriers**: High equipment costs, limited technical infrastructure, and the scarcity of trained personnel.**Ethical and legal challenges**: Issues related to digital privacy, admissibility of imaging evidence, algorithmic bias, and the need for standardization.**Interdisciplinary collaboration and training**: The importance of integrated efforts between radiologists, pathologists, and forensic scientists, and the call for updated education and role clarity.**Cultural adaptability and global collaboration**: The potential of non-invasive imaging techniques to facilitate forensic examinations in culturally sensitive or religiously conservative contexts, especially in the Middle East and Islamic countries.

These themes were not only descriptive but also interpretative, reflecting how structural and contextual factors intersect to shape the implementation and acceptance of imaging technologies in forensic medicine. Together, the narrative and thematic analyses provided a comprehensive synthesis of the literature, informed recommendations for practice, and highlighted priorities for future research and policy development in forensic imaging.

## 3. Results

### 3.1. Risk of Bias Assessment

The Risk of Bias (RoB) assessment for the included studies [16,34,35,36,37,38,39,40,41,42], highlights several critical areas requiring attention. Overall, the studies exhibit a moderate level of bias, with domains such as methodological rigor and data integrity being consistent strengths across most articles. However, notable concerns were identified in certain areas. For instance, several studies, including those by Ketsekioulafis et al. (2024) [35] and Tournois et al. (2024) [41], displayed “some concerns” in Domain 3 (D3), indicating potential issues with the reliability of outcome measurements or data reporting. Similarly, Patyal & Bhatia (2021) [38] and Suhas et al. (2024) [34] encountered challenges in Domain 2 (D2), highlighting the need for more robust procedural clarity and consistency in research design. The overarching risk in Domain 5 (D5), reflecting concerns around applicability and generalizability, was identified in studies like Chango et al., 2024 [16], emphasizing the need to better align findings with real-world forensic practices. Despite these concerns, foundational studies such as Beck (2011) [37] and Daly (2019) [36] maintained low overall risk, underscoring their methodological soundness and contribution to the field. The full RoB profile across all domains is summarized in Figure 2. Addressing these discrepancies through enhanced standardization, transparent reporting, and comprehensive validation protocols will be pivotal in ensuring the reliability and applicability of forensic research findings.

### 3.2. Main Outcomes

Based on the detailed data extracted from the included studies (Table 2), four primary outcome themes emerged, providing a comprehensive understanding of the integration, challenges, and future directions of imaging technologies and artificial intelligence (AI) applications in forensic medicine:

#### 3.2.1. Advancements in AI and Imaging Technologies

A central outcome consistently reported across the included studies was the enhancement of diagnostic precision and operational efficiency through advanced imaging modalities and AI-driven analytic tools.

Ketsekioulafis et al. (2024) [35] and Tournois et al. (2024) [41] demonstrated that AI algorithms trained on postmortem datasets significantly improved estimations of postmortem intervals and detection of trauma patterns compared to traditional methods, reducing subjective interpretation errors. Daly (2019) [36] emphasized the benefits of multi-detector computed tomography (MDCT) and 3D postmortem CT angiography (PMCTA), showing that these technologies allowed for highly detailed internal examinations without body dissection, preserving body integrity crucial for cultural and religious contexts.

Furthermore, Guglielmi et al. (2015) [40] highlighted that integrating virtual autopsy techniques into medico-legal practices led to improved injury visualization and the possibility of case review over time, a feature not possible with traditional autopsy. Despite these advancements, several studies, including those by Patyal & Bhatia (2023) [38], noted limitations in AI model generalizability due to a lack of diverse forensic datasets, underscoring the need for ongoing algorithm refinement.

#### 3.2.2. Operational and Financial Barriers

A significant theme across the studies was the challenge posed by operational costs and financial limitations in adopting advanced forensic imaging technologies.

Beck (2011) [37] and Chango et al., 2024 [16], reported that the acquisition and maintenance of CT and MRI machines impose substantial financial burdens on forensic institutions, particularly in low- and middle-income countries. Operational costs such as electricity, software maintenance, and the need for highly skilled operators further exacerbate resource disparities between high-income and resource-limited settings.

Innovative approaches to cost reduction were discussed, including the use of portable imaging devices and mobile IoT-driven scanners for remote forensic investigations, as highlighted by Chango et al., 2024 [16]. However, even these solutions face hurdles related to device calibration, data transmission security, and specialized training. These findings emphasize the urgent need for scalable, cost-effective, and decentralized forensic imaging models that maintain high diagnostic fidelity while remaining financially sustainable.

#### 3.2.3. Ethical and Legal Considerations

The integration of imaging and AI technologies into forensic workflows introduces complex ethical and legal challenges that were systematically discussed across several studies.

Suhas et al. (2024) [34] underscored concerns regarding data security and digital postmortem privacy, highlighting the vulnerability of sensitive imaging data to breaches and unauthorized access if cybersecurity protocols are inadequate. Blockchain-based solutions were proposed to safeguard chain-of-custody processes and ensure data integrity throughout forensic workflows.

Guglielmi et al. (2015) [40] and Daly (2019) [36] emphasized the lack of standardized protocols governing medico-legal imaging, noting that inconsistencies in acquisition methods and interpretation frameworks could compromise the admissibility of forensic imaging evidence in court proceedings. The studies collectively stressed the necessity of developing universal guidelines for forensic imaging, encompassing technical standards, data storage policies, and ethical governance, to enhance both the scientific validity and legal credibility of forensic imaging findings.

#### 3.2.4. Interdisciplinary Collaboration, Training, and Cultural Adaptation

A final emergent theme was the critical importance of interdisciplinary collaboration, structured training, and cultural adaptability in successfully integrating advanced imaging technologies into forensic practice.

Patyal & Bhatia (2023) [38] and Malfroy Camine et al. (2024) [39] demonstrated that effective forensic imaging requires seamless collaboration between forensic pathologists, radiologists, anthropologists, and data scientists. Studies revealed that role ambiguity, lack of mutual understanding of discipline-specific challenges, and absence of standardized interdisciplinary workflows hindered optimal integration of imaging into forensic processes.

Training programs specifically addressing both imaging interpretation and forensic principles were proposed as necessary for bridging skill gaps. Furthermore, Daly (2019) [36] and Guglielmi et al. (2015) [40] highlighted the transformative role of virtual autopsies in regions where traditional autopsies face religious or cultural objections, particularly in Middle Eastern and Islamic contexts. By offering non-invasive alternatives, imaging technologies were seen not only as diagnostic tools but also as facilitators of culturally respectful forensic practices.

## 4. Discussion

This systematic review underscores the transformative potential of integrating advanced imaging technologies and artificial intelligence (AI) in forensic medicine. The analyzed literature clearly demonstrates that AI-enhanced imaging tools—such as virtual autopsy (virtopsy), multi-detector computed tomography (MDCT), and AI-driven diagnostic algorithms—significantly improve diagnostic accuracy, reproducibility, and operational efficiency in forensic investigations [26,43]. Despite these clear benefits, the review identified critical operational, ethical, and interdisciplinary challenges that must be strategically addressed to facilitate broader global adoption [44].

A core finding of this review is the remarkable improvement in forensic diagnostic precision and reproducibility provided by AI-powered imaging. AI tools such as machine learning algorithms effectively reduce human error and variability in trauma interpretation, fracture detection, and postmortem interval estimations [45,46]. However, to ensure these improvements translate into reliable judicial outcomes, it is crucial that imaging findings meet rigorous standards for legal admissibility [47]. Forensic evidence must not only be scientifically sound but also be consistently reproducible, transparent, and defensible under cross-examination in court [48]. Thus, accuracy and evidentiary reliability significantly outweigh considerations of implementation costs, as noted by multiple studies [49]. While financial barriers are genuine concerns, the ultimate priority must be ensuring the scientific robustness and legal validity of forensic imaging results [50].

This review also reveals an important yet underexplored opportunity for innovation through the establishment of novel international forensic collaboration models leveraging AI-based imaging technologies. Given the growing demand for culturally respectful, non-invasive autopsy alternatives—particularly in Middle Eastern and Islamic contexts—international cooperation in forensic imaging can be both ethically advantageous and operationally beneficial [51]. A proposed “Global Virtual Forensic Network” (GVFN) could be established, connecting forensic centers worldwide through secure blockchain-supported digital imaging repositories [52,53]. This network would enable secure, real-time consultation among international forensic experts, thereby facilitating culturally sensitive examinations, joint diagnostics, and standardized training programs. Such collaborative models could dramatically enhance global forensic standards, expand diagnostic capabilities, and improve equitable access to forensic resources in low-resource regions [54].

Ethical and considerations emerged prominently within the review and require robust attention in future forensic imaging practices [9]. The digitization and remote transfer of highly sensitive postmortem imaging data present significant privacy risks and ethical challenges, necessitating stringent data governance measures [55,56]. Blockchain technologies, suggested by several included studies, offer promising solutions for safeguarding chain-of-custody integrity, protecting data privacy, and mitigating cyber vulnerabilities [57]. Ethical compliance is equally critical for addressing potential algorithmic biases inherent in AI applications, which could compromise the neutrality and objectivity required by forensic investigations [58]. Ensuring transparency, traceability, and fairness in AI processes will not only strengthen judicial confidence but also promote broader societal acceptance of AI-based forensic evidence.

Moreover, the implementation of imaging technologies in forensic medicine necessitates structured interdisciplinary collaboration and education frameworks. Radiologists, forensic pathologists, anthropologists, and data scientists must engage collaboratively, supported by training curricula that bridge technical and forensic disciplines [59]. Educational programs should incorporate training modules on AI application, interpretation of imaging-based forensic evidence, and adherence to ethical guidelines. Such initiatives would reduce inter-professional knowledge gaps and foster cohesive interdisciplinary forensic teams capable of effectively utilizing these advanced imaging tools [60,61].

While financial and resource constraints were identified as critical barriers, the review emphasizes that accuracy, reliability, and legal admissibility of forensic findings must be prioritized over cost considerations. Indeed, operational expenses associated with advanced imaging technologies remain substantial; however, the societal and judicial implications of inaccurate forensic diagnoses are significantly greater, reinforcing the necessity of investments in high-quality forensic imaging capabilities [62,63]. Potential solutions include adopting portable imaging technologies and shared-resource models between forensic and healthcare institutions, thereby balancing financial sustainability with diagnostic accuracy [64].

### 4.1. Implications

The findings of this systematic review emphasize the transformative potential of advanced imaging technologies in forensic medicine. These technologies, particularly AI-driven imaging, virtual autopsies, and IoT-enabled tools, promise to revolutionize forensic investigations by enhancing diagnostic accuracy, reducing subjectivity, and streamlining workflows. From a practical perspective, integrating such technologies can improve the timeliness and reliability of forensic analyses, which is critical in criminal investigations and legal proceedings. Furthermore, adopting non-invasive imaging techniques can address cultural and religious sensitivity surrounding traditional autopsies, thereby increasing public trust in forensic systems. On a policy level, the development of standardized protocols for acquiring, storing, and presenting imaging data is vital for ensuring legal admissibility and evidentiary value. Collaborative efforts among international forensic organizations, such as INTERPOL and ENFSI, can foster the establishment of universal guidelines and best practices. Additionally, investments in infrastructure and training programs are necessary to bridge the gap in resource-limited settings, ensuring equitable access to these advanced diagnostic tools. Finally, interdisciplinary collaboration between forensic pathologists, radiologists, and technologists must be prioritized to maximize the potential of these innovations in forensic practice.

### 4.2. Critical Appraisal of Included Studies

While the reviewed studies collectively demonstrate the promising role of advanced imaging and artificial intelligence (AI) in forensic diagnostics, several methodological limitations merit closer scrutiny. A recurrent issue across the dataset is the limited generalizability of findings due to sample homogeneity or context-specific constraints. For instance, Ketsekioulafis et al. [35] and Patyal & Bhatia [38] conducted systematic reviews that heavily relied on datasets derived from high-income regions, thereby underrepresenting forensic scenarios in low-resource or culturally diverse environments. This raises concerns about the universal applicability of their conclusions, particularly regarding AI model performance, which is highly sensitive to training data variability.

Additionally, several studies, such as Suhas et al. (2024) [34] and Tournois et al. (2024) [41], exhibited moderate risk of bias in domains related to procedural transparency and outcome measurement reliability. In some cases, small sample sizes or reliance on pilot data—often derived from specialized forensic centers—limit statistical robustness and hinder the extrapolation of results to broader medico-legal contexts. Moreover, while ethical and legal implications were mentioned across many studies, only a few provided detailed frameworks for how such challenges were operationalized or addressed, highlighting a need for more practice-oriented empirical research.

To enhance future evidence synthesis, there is a pressing need for multicenter studies encompassing varied demographic, cultural, and resource settings. This would not only improve external validity but also support the development of AI models and imaging protocols that are resilient across forensic populations and legal systems worldwide.

### 4.3. Limitations

While this review provides valuable insights into the role of advanced imaging technologies in forensic medicine, it is not without limitations. First, the included studies varied in design and quality, with some relying on small sample sizes or focusing on specific case studies, which may limit the generalizability of the findings. Second, most studies were conducted in high-resource settings, with limited data from low- and middle-income countries, where resource constraints may significantly impact the feasibility of adopting these technologies. Third, the rapid pace of technological advancements in forensic imaging means that some recent developments may not have been fully captured within the review’s timeframe. Additionally, many studies did not comprehensively address the potential biases in AI algorithms and the lack of diverse datasets, which could influence the reported outcomes. Lastly, while discussed, ethical and legal considerations remain underexplored in practical implementation, requiring further research and clarification.

## 5. Conclusions

Advanced imaging technologies represent a significant leap forward in the field of forensic medicine, offering enhanced diagnostic accuracy, non-invasive investigative options, and improved documentation of evidence. Despite their potential, the successful implementation of these technologies requires addressing several operational, ethical, and legal challenges. Standardized protocols, robust training programs, and interdisciplinary collaboration will be key to overcoming these barriers. This review underscores the need for continued research, particularly in low-resource settings, to ensure these advancements’ global applicability and equitable distribution. By addressing these gaps, forensic medicine can fully leverage the transformative power of imaging technologies to improve the accuracy, efficiency, and reliability of forensic investigations.

## Figures and Tables

**Figure 1 diagnostics-15-01410-f001:**
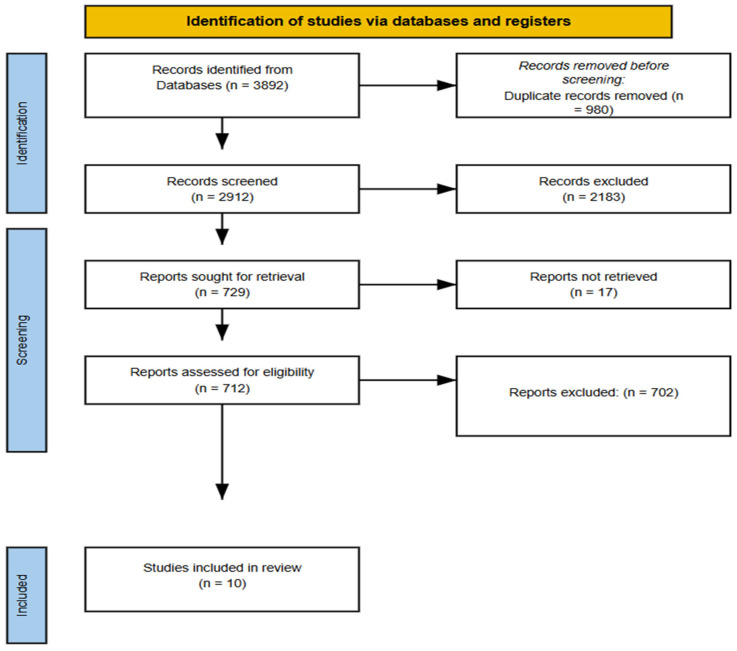
PRISMA flow diagram [16,34,35,36,37,38,39,40,41,42].

**Figure 2 diagnostics-15-01410-f002:**
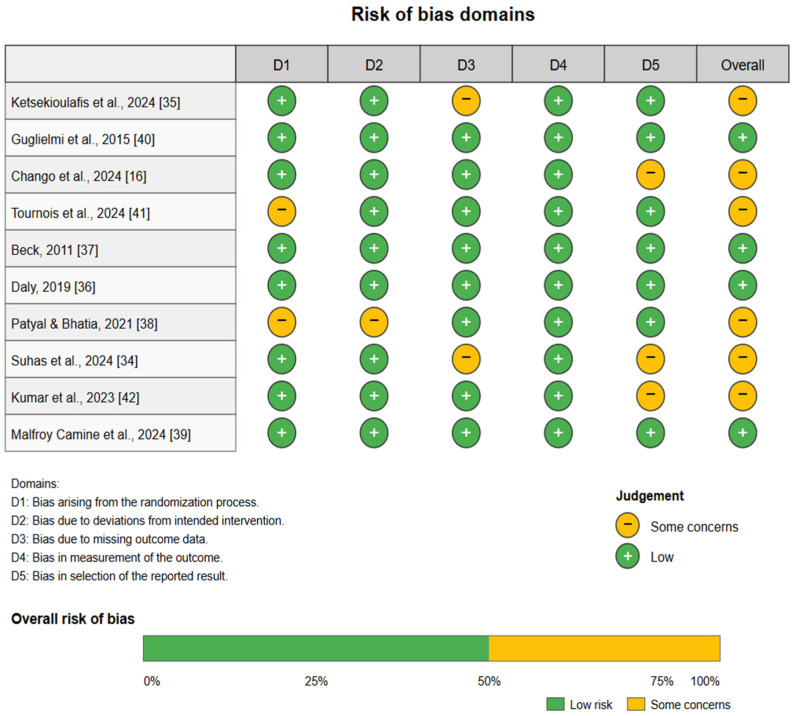
Risk of bias assessment [16,34,35,36,37,38,39,40,41,42].

**Table 1 diagnostics-15-01410-t001:** Search Strategy.

Database	Search Terms
**PubMed**	(“Forensic Medicine”[Mesh] OR “Post-mortem Imaging” OR “Virtopsy”) AND (“CT Scan” OR “MRI” OR “Virtual Autopsy”) AND (“Artificial Intelligence” OR “Machine Learning”) AND (“Challenges” OR “Ethical Issues” OR “Operational Barriers”)
**Scopus**	TITLE-ABS-KEY (“forensic medicine” OR “postmortem imaging” OR “virtual autopsy”) AND TITLE-ABS-KEY (“computed tomography” OR “magnetic resonance imaging”) AND TITLE-ABS-KEY (“AI” OR “machine learning” OR “technological innovations”) AND TITLE-ABS-KEY (“challenges” OR “ethical considerations” OR “implementation barriers”)
**Web of Science**	TS = ((“forensic medicine” OR “postmortem imaging” OR “virtual autopsy”) AND (“CT” OR “MRI” OR “3D imaging”) AND (“artificial intelligence” OR “machine learning”) AND (“challenges” OR “ethical considerations” OR “training needs”))
**CINAHL**	(“Forensic Medicine” OR “Postmortem Radiology”) AND (“Artificial Intelligence” OR “Machine Learning”) AND (“Diagnostic Accuracy” OR “Operational Challenges” OR “Implementation Barriers”)
**Embase**	(‘forensic medicine’/exp OR ‘forensic imaging’/exp) AND (‘medical imaging’/exp OR ‘computed tomography’/exp OR ‘magnetic resonance imaging’/exp) AND (‘artificial intelligence’/exp OR ‘machine learning’) AND (‘ethical issues’ OR ‘operational barriers’)
**ProQuest**	(“Forensic Imaging” OR “Virtual Autopsy” OR “Medical Imaging in Forensics”) AND (“CT Scan” OR “MRI”) AND (“AI applications” OR “Machine Learning”) AND (“Ethical Challenges” OR “Implementation Barriers”)
**Elsevier/ScienceDirect**	(“Forensic Medicine” OR “Postmortem Imaging”) AND (“Computed Tomography” OR “MRI”) AND (“Artificial Intelligence” OR “Machine Learning”) AND (“Operational Challenges” OR “Ethical Issues”)

**Table 2 diagnostics-15-01410-t002:** Data Extraction for Systematic Review.

Author(s), Year	Study Design	Sample Size	Setting	Imaging Technology and Innovation	Challenges and Limitations	Outcome Measures	Key Findings
Ketsekioulafis et al., 2024 [35]	Systematic Review	33 articles	Forensic Sciences Settings	AI for postmortem interval estimation, cause of death	Algorithmic bias, data dependency	Enhanced diagnostic support	AI enhances forensic diagnostics; requires further validation
Guglielmi et al., 2015 [40]	Editorial Review	N/A	Radiology and Forensic Labs	MDCT, postmortem imaging	Integration into medico-legal systems	Improved autopsy visualization	Advanced imaging aids forensic investigations; requires legal and ethical guidelines
Chango et al., 2024 [16]	Systematic Review	63 articles	Forensic Science Investigations	Extended reality, biometric data analysis, AI techniques	Ethical and operational challenges	Improved crime scene analysis	Technology enhances precision and efficiency; implementation challenges identified
Tournois et al., 2024 [41]	Scoping Review	35 articles	Medicolegal Practice	AI for biological age estimation, postmortem analysis	Limited practical integration	AI maturity levels	AI is mostly in the R&D stages; it highlights the potential for medicolegal applications
Beck, 2011 [37]	Narrative Review	N/A	Forensic Imaging Practices	Radiography, CT, MRI	Cost, data security	Forensic imaging utility	Reviews radiographic evolution; discusses imaging challenges and future prospects
Daly, 2019 [36]	Editorial Review	N/A	Forensic Radiology Settings	3D postmortem CT, PMCTA	Public and cultural opposition to autopsy	Enhanced postmortem documentation	Non-invasive imaging improves forensic reporting; addresses cultural sensitivities
Patyal & Bhatia, 2021 [38]	Systematic Review	20 studies	Radio-Diagnostic Modalities	AI and ML in forensic radiology	Training, ethical issues	Enhanced diagnostic accuracy	AI facilitates forensic radiology; training gaps and ethical concerns are highlighted
Suhas et al., 2024 [34]	Review Article	N/A	Digital Forensic Analysis	Blockchain, digital imaging, 3D modeling	Data security, high costs	Improved evidence integrity	Digital tech enhances forensic precision; blockchain ensures data integrity
Kumar et al., 2023 [42]	Conference Proceedings/Review	N/A	Medical Imaging Systems	AI-based medical imaging systems, deep learning applications	Computational complexity, data privacy, integration challenges	Future directions for AI implementation	Identifies key challenges in AI-based imaging; proposes future research directions
Malfroy Camine et al., 2024 [39]	Case Study	N/A	Forensic Identification Cases	Virtual re-association (VRA), postmortem CT	Fragmented data challenges	Enhanced identification accuracy	VRA supports human remains identification in complex cases

## Data Availability

Data will be available upon request.

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
