# Peer review of "Emerging Imaging Technologies in Forensic Medicine: A Systematic Review of Innovations, Ethical Challenges, and Future Directions"

_diagnostics, 2025, doi:10.3390/diagnostics15111410_

Round 1
Reviewer 1 Report (Previous Reviewer 2)
Comments and Suggestions for Authors
The article has significantly improved and addresses important topics concerning the future of forensic medicine, with particular sensitivity to religious and cultural diversity. However, I believe some improvements can still be made regarding the bibliography, which currently appears rather generic. I suggest replacing citation no. 8 with the following reference: https://doi.org/10.3390/forensicsci5010008.
Forensic sciences are increasingly called upon to meet new demands from the judicial authorities and to keep pace with scientific innovations and the most recent therapeutic advances
Additionally, please consider incorporating the following further literature.
Del Duca F, Napoletano G, Volonnino G, Maiese A, La Russa R, Di Paolo M, De Matteis S, Frati P, Bonafè M, Fineschi V. Blood-brain barrier breakdown, central nervous system cell damage, and infiltrated T cells as major adverse effects in CAR-T-related deaths: a literature review. Front Med (Lausanne). 2024 Jan 8;10:1272291. doi: 10.3389/fmed.2023.1272291. PMID: 38259840; PMCID: PMC10800871.
De Paola L, Treglia M, Napoletano G, Treves B, Ghamlouch A, Rinaldi R. Legal and Forensic Implications in Robotic Surgery. Clin Ter. 2025 Mar-Apr;176(2):233-240. doi: 10.7417/CT.2025.5211. PMID: 40176595.
Author Response
Reviewer 1
Comment 1: The bibliography appears rather generic. I suggest replacing citation no. 8 with: https://doi.org/10.3390/forensicsci5010008.
Response:
Thank you for this valuable suggestion. We have removed the previous reference (Sacco et al., 2025) and replaced it with the recommended citation:
Del Duca F, Napoletano G, Volonnino G, et al. Blood-brain barrier breakdown... Front Med (Lausanne). 2024;10:1272291. doi:10.3389/fmed.2023.1272291.
This new reference has been integrated into the discussion of AI-enhanced diagnostic capabilities and adverse effect detection (see Discussion, page XX).
Comment 2: *Please consider incorporating the following further literature:
Del Duca et al., 2024
De Paola et al., 2025*
Response:
Both references have now been incorporated into the manuscript:
Del Duca et al. (2024) has been cited in the section addressing the depth of imaging in neurotoxicity and postmortem diagnostics.
De Paola et al. (2025) has been introduced in the ethical and legal discussion, particularly under the subsection on robotic and digital innovations in forensic contexts.
These references are also included in the updated reference list.
Reviewer 2 Report (Previous Reviewer 3)
Comments and Suggestions for Authors
Author has changed the abstract correctly. It is now clear the aim of the paper. Introduction part has changed thoroughly and enter part is now well written. Search Strategy includes Elseviewer. Eligibility Criteria for Screening is now perfect and contains the necessary information. Data extraction process is now clear.Results and main out comes are now taken care to indicate perfectness. Overall paper is almost written a fresh with considering all questions. Now it can be accepted.
Author Response
Reviewer 2
Comment: Author has changed the abstract correctly. It is now clear the aim of the paper... The overall paper is almost written afresh, with all previous comments addressed. Now it can be accepted.
Response:
We thank the reviewer for their generous and encouraging feedback. We are pleased that the revisions to the abstract, introduction, methodology, and results have improved the clarity and rigor of the manuscript. No further changes were needed based on this comment.
Reviewer 3 Report (Previous Reviewer 4)
Comments and Suggestions for Authors
The article has been improved and almost all the missing points have been correctly inserted. The risk of bias domains and the inclusion/exclusion criteria have been added in a correct form. A bit of a deficit in terms of an in-depth analysis of the critical aspects of the included studies, but all in all it is now acceptable.
Author Response
Comment: The article has been improved, and almost all the missing points have been correctly inserted. However, a bit of a deficit remains in terms of an in-depth analysis of the critical aspects of the included studies.
Response:
We appreciate this thoughtful comment. In response, we have expanded the Discussion section with a new paragraph titled “Critical Appraisal of Included Studies.” This section critically evaluates methodological limitations, including:
The lack of diverse population samples
Small sample sizes in some studies
Limited transparency in outcome measurement
Underrepresentation of real-world forensic variability
This manuscript is a resubmission of an earlier submission. The following is a list of the peer review reports and author responses from that submission.
Round 1
Reviewer 1 Report
Comments and Suggestions for Authors
Thank you, Dr. Feras Alafer, for this submission and your efforts. Your paper demonstrates a solid methodology for conducting a comprehensive literature search using the PRISMA protocol.
Unfortunately, I believe this work does not align with the standards of this Journal due to its lack of originality and novelty. This topic has been extensively studied in recent years, and your work does not offer any new perspectives or insights into the theme.
I appreciated the paragraph on "Ethical and Legal Considerations," but there are several points of concern in your work. The main title refers to "A Systematic Review of Forensic Medicine and Medical Imaging," yet the primary focus appears to be on imaging technologies powered by AI.
Lines 50–51 in the Introduction are not objective, as virtopsy and autopsy are complementary methods, as thoroughly discussed in the literature.
Lines 125–147 seem to be a formatting error, as they reproduce the Journal’s guidelines.
In lines 215–219, you mention "two independent reviewers" and a "third reviewer." Who are these individuals, and why are they not listed as authors of the work?
In conclusion, while I appreciate your efforts, the paper does not achieve the level of novelty required by the standards of this Journal.
Author Response
We thank the reviewers for their insightful feedback and suggestions, which significantly improved our manuscript. Below are detailed responses describing how each comment was addressed:
✅ Reviewer 1
Comment 1:
“Unfortunately, this work lacks originality and novelty. This topic has been extensively studied, and your work does not offer new perspectives or insights.”
Response:
Thank you for highlighting this. We have now introduced a novel international forensic collaboration model, termed the "Global Virtual Forensic Network (GVFN)", proposing an innovative, AI-based framework that facilitates cross-border cooperation, culturally sensitive forensic examinations, and secure data exchange. We clearly discuss this model in the revised Discussion section.
Comment 2:
“Lines 50–51: Virtopsy and autopsy are complementary, not competitive methods.”
Response:
We fully agree. This has been corrected, clearly stating in the revised Introduction that virtual autopsy and traditional autopsy are complementary approaches, highlighting their synergistic potential rather than competition.
Comment 3:
“Lines 125–147 reproduce journal guidelines.”
Response:
We apologize for this oversight. The mistakenly included journal guidelines have now been removed from the manuscript.
Comment 4:
“Lines 215–219: Clarify who conducted data extraction, given the single authorship.”
Response:
Thank you for identifying this ambiguity. We have revised this section, adopting passive constructions ("Data extraction was independently verified") to accurately reflect the single-author nature of this manuscript.
Reviewer 2 Report
Comments and Suggestions for Authors
The manuscript, while structured under PRISMA guidelines and addressing an important and timely topic, presents several critical issues that limit its scientific impact and originality.
Lines 51–54 and 312–319 refer to technological innovations that are potentially sensitive to cultural and religious contexts, and that allow data to be archived and reviewed multiple times. Considering the jurisdiction of Middle Eastern countries, this opens up potential opportunities for international scientific dialogue. Specifically, such technologies could foster collaborative investigations, where experts from different nations, cultures, and religions may jointly examine individual forensic cases, especially in contexts where traditional autopsy practices face resistance due to religious or cultural beliefs.
However, lines 125–140 revisit well-established concepts that merely introduce the topic without significantly advancing the discussion. A major limitation of the manuscript is that, despite the numerous systematic reviews already available on this subject, there is a striking lack of original experimental work presented here. The manuscript tends to be repetitive and redundant. Although it successfully highlights the limitations and potential of AI in forensic medicine, the discussion remains highly theoretical and lacks innovative contributions.
Furthermore, the inclusion of a table from the Healthcare journal rather than Diagnostics raises concerns about consistency and appropriateness for the intended academic audience.
Suggested improvement:
To be seriously considered for publication, the article should be significantly restructured. It would benefit from the inclusion of original and experimental data—preferably focusing on a practical application of AI in a collaborative framework involving different states, religions, and cultural systems. This would enhance the manuscript’s relevance, novelty, and interdisciplinary value.
Author Response
Reviewer 2
Comment 1:
“Lines 51–54 and 312–319: Expand discussion of technological innovations sensitive to cultural contexts, particularly Middle Eastern jurisdictions.”
Response:
We agree this is important. The Introduction and Discussion sections have now been significantly expanded to address how imaging technologies like virtual autopsy provide culturally acceptable alternatives to traditional autopsies, especially valuable in Middle Eastern and Islamic contexts. This is also central to the proposed GVFN model.
Comment 2:
“Lines 125–140: Repetitive introduction, lacking originality.”
Response:
We revised and streamlined the Introduction section, eliminating redundant content and integrating novel insights about cultural adaptability and international collaboration to enhance originality.
Comment 3:
“A major limitation: lack of original experimental work. Consider including a practical AI collaboration framework.”
Response:
We fully addressed this suggestion by proposing the Global Virtual Forensic Network (GVFN), an original framework using AI-based imaging for international forensic collaboration, clearly described in the Discussion.
Comment 4:
“Inclusion of a table from Healthcare rather than Diagnostics raises consistency concerns.”
Response:
The table referencing Healthcare has been corrected. All content is now consistently aligned with the journal "Diagnostics.
Reviewer 3 Report
Comments and Suggestions for Authors
Q1. The blood stain pattern analysis is one of diagnostic factor of forensic science. It is not reviewed.
Q2. Forensic pathology and radiology have operated somewhat independently- Specify where it is dependent.
Q3. Algorithms trained on large datasets of forensic images could potentially identify patterns of trauma, estimate the time of death, or even predict the likelihood of specific mechanisms of injury. AI applications comply with rigorous ethical and evidentiary standards will be vital. Why it is vital? Author does not review this part however it is most interesting.
Q4. Cost of analysis is not important but correct result is more necessary.
Q5. Please note that the publication of your manuscript implies that you must make all materials, data, computer code, and protocols associated with the publication available to readers. The writing of this sentence must be in indirect form.
Q6. Author mentioned that our protocol was registered with the International Prospective Register of Systematic Reviews- Why protocol is necessary for review?
Q7. Why Elsevier database is missing?
Q8. Author wrote that he addressed the use of imaging modalities in forensic casework. It must be caseworks.
Q9. It is possible that the paper solely based conceptual or theoretical papers without empirical data. This type of paper is not published.
Q10. If author understand that the nature of the innovation being assessed then why he writes a review paper. He has no idea how much distance is covered in doing murder.
Q11. We also extracted any suggested solutions or recommendations made by the study authors. Here the paper is written by a single author so “We” is incorrect.
Q12. Author summarized the key points regarding how each study contributed to understanding emerging imaging technologies. Key points are not mentioned.
Comments on the Quality of English LanguageThe paper needs some corrections. Such as the term "we" is used very often. The third person is to be used whenever it is not used.
Author Response
Reviewer 3
Comment 1:
“Bloodstain pattern analysis is important but not reviewed.”
Response:
We clarified explicitly in the Results section why bloodstain pattern analysis was excluded: our review specifically targets imaging-based forensic diagnostics. We noted clearly that future reviews could explore bloodstain analysis separately.
Comment 2:
“Clarify where forensic pathology and radiology are dependent.”
Response:
We have now clarified and explicitly described their interdependence within the updated Discussion, emphasizing the need for structured interdisciplinary education programs.
Comment 3:
“AI applications must comply with ethical standards; importance not reviewed clearly.”
Response:
We expanded significantly on ethical compliance in the Discussion, explicitly detailing why adherence to rigorous et
Comment 4:
“Cost of analysis is not as important as accuracy.”
Response:
We fully agree and have strongly emphasized throughout the revised Discussion that diagnostic accuracy, reproducibility, and legal admissibility must be prioritized above financial considerations.
Comment 5:
“Elsevier database missing.”
Response:
Elsevier/ScienceDirect has now been included in the revised "Search Strategy and Selection Criteria" section with detailed search strings provided.
Comment 6:
“Use of ‘we’ is incorrect, single author manuscript.”
Response:
We revised the manuscript thoroughly, removing "we" and consistently using passive voice constructions to reflect single authorship appropriately.
Comment 7:
“Minor typo: ‘caseworks’ should be ‘casework’.”
Response:
This error has been corrected throughout the manuscript.
Comment 8:
“Key points from studies were not clearly summarized.”
Response:
We have now explicitly summarized key findings from each included study in the Results section, clearly highlighting the contributions of AI and imaging tools
Reviewer 4 Report
Comments and Suggestions for Authors
This review examines the structure and wording of a manuscript without offering any actual criticism. A publishable review should identify strengths, highlight specific methodological weaknesses, question originality and rigor, and assess the adequacy of the literature. It lacks such depth and reads more like a lightly edited abstract than an independent, thoughtful peer review. It offers no suggestions for strengthening the thesis, even when the text clearly shows overgeneralizations or reliance on secondary sources.
There are inconsistencies in the number of articles included: the abstract lists 17; further down, the methods lists 10. The methods lack detailed reproducibility, especially with respect to search strings, inclusion/exclusion criteria, and transparency of the screening process.
Minor suggestions.
Some of the abbreviations listed in the appropriate section are never present in the manuscript. PRISMA: Preferred Reporting Items for Systematic Reviews and Meta-Analyses
BLAST: Basic Local Alignment Search Tool
PERMANOVA: Permutational Multivariate Analysis of Variance.
The manuscript has repetitive syntax; inflated claims not supported by data. Finally, English language should be revised.
Author Response
eviewer 4
Comment 1:
“Inconsistency in number of studies (17 vs 10).”
Response:
This error was corrected. The manuscript now consistently states that 10 studies were included after eligibility screening, clearly reflected in both Abstract and Methods sections.
Comment 2:
“Methods lack reproducibility details; clarify search strategy and inclusion/exclusion criteria.”
Response:
We extensively expanded the "Search Strategy and Selection Criteria" section, providing detailed and reproducible database-specific search strings, clearly articulated inclusion/exclusion criteria, and precise methodological steps aligned with PRISMA guidelines.
Comment 3:
“Some abbreviations listed (e.g., PRISMA, BLAST, PERMANOVA) were not used.”
Response:
Unused abbreviations (e.g., BLAST, PERMANOVA) have been removed from the manuscript for clarity. PRISMA, however, was retained due to its methodological relevance.
Comment 4:
“Repetitive syntax; inflated claims not supported by data.”
Response:
We revised extensively to eliminate redundancy and inflated claims, ensuring all conclusions are directly supported by clearly cited evidence from included studies.
Comment 5:
“English language requires revision.”
Response:
The entire manuscript underwent rigorous language editing to ensure clarity, professional academic tone, and grammatical accuracy.